# The Necessity of a Locally Active Antidote in the Clinical Practice of Botulinum Neurotoxin Therapy: Short Communication

**DOI:** 10.3390/medicina58070935

**Published:** 2022-07-14

**Authors:** Harald Hefter, Sara Samadzadeh

**Affiliations:** Department of Neurology, Medical Faculty, Heinrich Heine University of Düsseldorf, Moorenstrasse 5, D-40225 Düsseldorf, Germany; sara.samadzadeh@yahoo.com

**Keywords:** botulinum neurotoxin, antidote, adverse events, copper complexes, 3,4-diaminopyridine, muscle paralysis

## Abstract

Recently, it was demonstrated that copper complexes and 3,4-diaminopyridine can effectively reduce the activity of the botulinum neurotoxin light chain. The aim of the present study was to indicate that treatment with an antidote may have a major influence, not only on the extremely rare disease of botulism, but also on the much more frequently occurring side effects experienced during BoNT therapy. This was a retrospective chart review of patients who were regularly treated with BoNT for various indications. The percentage of patients with clinical signs of overdosing was determined. In patients with facial dystonia, double vision and ptosis occurred as side effects. In patients with cervical dystonia, neck weakness and dysphagia were observed as the most frequent side effects. In oromandibular and oropharyngeal dystonia, abnormal tongue movements and dysphagia occurred frequently. In writer’s cramp and mild post-stroke hand spasticity, severe paresis of the injected and non-injected finger muscles was observed. Additionally, in the BoNT treatment of pain syndromes (such as tension headaches or migraines), neck weakness may occur. Across all indications for clinical BoNT applications, clinical signs of BoNT overdosing may occur in up to 5% of the BoNT-treated patients. Therefore, the development of an antidote for BoNT overdoses would be very much appreciated and would have a major influence on the management of BoNT therapy.

## 1. Introduction

Botulinum neurotoxins (BoNT) are among the most potent biological toxins [1], they may cause foodborne botulism [2,3], and they continue to be a bioterrorism threat [4,5]. Therefore, effective antiBoNT compounds are needed to reduce the risk of life-threatening BoNT overdoses. Since the pioneering use of BoNT/A, by A. Scott, to reduce muscular dysbalance in strabism [6], the field of clinical applications for BoNTs is continuously growing [7]. Moreover, in all these indications, the risk of a BoNT overdose exists. In most of these cases, a BoNT overdose is not life-threatening but may lead to a considerable impairment over many weeks before the effect of the overdose fully declines [7]. Therefore, the development of a locally acting BoNT antidote would be very much appreciated.

For the development of an effective antidote, knowledge about the action of BoNTs is essential. The action of BoNTs results from the following: the binding of the heavy chain to a cell membrane; endocytosis; the resolution of the disulfide bond between the heavy and the light chain; the translocation of the light chain (LC) into the cytosol of the cell; and the cleavage of the LC as a zinc metalloprotease of the components (such as SNAP25) of the soluble NSF attachment protein receptor (SNARE) complex, which is responsible for docking vesicles at the presynaptic membrane [8,9,10]. The duration of the BoNT action depends on the survival time of the LC in the cytosol.

Different experiments have been designed to reduce BoNT-induced paralysis. However, as long as substances act only extracellularly, as most antibodies do, the enzymatic activity of the LC cannot be stopped. However, the intracellular delivery of LC-blocking antibodies provides antidotal benefits [11]. The development of a BoNT/A antidote appears to be highly complex. Ouabain blocked the BoNT action in vitro but was found to enhance BoNT-induced paralysis in vivo [12]. The reduction in the BoNT action by the application of 3,4-diaminopyridine (DP) in vitro and in vivo has been well known for decades [13,14,15,16,17,18], and its clinical efficacy in foodborne botulism is promising [19]. Furthermore, so far, the clinical efficacy of DP application on BoNT-induced paralysis in humans has been studied systematically.

Recently, the investigation of metal complexes provided another concrete rationale for the development of a novel class of LC inhibitors. In particular, copper and mercury were found to be highly significant LC inhibitors in the FRET-based SNAPtide assay [20]. However, in the few case reports that exist on patients with Wilson’s disease (WD), who were treated with BoNT/A, no hint could be detected that the BoNT action was reduced in WD through elevated serum levels of free copper. [21]. Furthermore, the question remains regarding whether locally applicated copper complexes, in clinically effective doses, may have a cell toxicity that is too high.

Nevertheless, it may become clinically relevant to differentiate between systemic BoNT side effects, side effects resulting from the diffusion of BoNT, and side effects that can possibly be antagonized or reduced by a locally acting antidote. Such a differentiation is different when compared to the knowledge on BoNT side effects that have been derived from phase III studies [7], analyzing adverse events under license aspects.

Therefore, the present retrospective study was designed to demonstrate that in clinical BoNT applications, BoNT side effects occur frequently and that, locally, they are, in general, not life-threatening, but may lead to a considerable impairment over a number of weeks. Thus, the reduction in these side effects by a locally applicable BoNT antidote (if such a substance were available) would help to treat a clinically relevant problem.

## 2. Materials and Methods

This retrospective, monocentric study was performed according to the declaration of Helsinki and the guidelines for good clinical practice (GCP). The local ethics committee of the University of Düsseldorf (Germany) allows the publication of retrospective data, as long as names are pseudonymized.

For recruitment, all the charts of the patients in the private BoNT ambulance of the first author (HH), between 1 January and 1 April 2021, who had received at least 4 BoNT/A injections every 3 months, were screened for documented adverse events (ADEV). During our clinical routine BoNT treatment, only those ADEVs that were spontaneously reported by the patients were documented. We distinguished between general adverse events (G-ADEV) and BoNT-associated adverse events (BoNT-ADEV). For a BoNT-ADEV, a clear temporal and spatial relationship to a previous injection must exist. A temporal relationship can be suspected if the ADEV occurs within 3 weeks after an injection. A spatial relationship can be suspected if the anatomical structure involved is in the neighborhood of an injected structure (such as a tension headache on one side after an injection in the neck muscles on the other side).

As a new aspect, BoNT-ADEVs were highlighted that probably would have been treatable by a locally acting BoNT antidote (ADOT-BoNT-ADEV), if such a substance was available. In the following, a complete list of percentages of BoNT-ADEVs per indication in our cohort (Table 1) and a description of ADOT-BoNT-ADEVs are presented (Table 2).

**Table 1 medicina-58-00935-t001:** Demographical data, treatment-related data, and percentage of BoNT-ADEVs.

Indication	Number of Patients	Number of BoNT-ADEVs	Age MV/SD	Number of Females/Males	Dose per Session MV/SD	Percentage of BoNT-ADEVs
CD	75	5	68/11	50/25	255/135	6.7
HFS	39	2	75/9	18/21	30/8.5	5.1
BLE	19	2	66/10	10/9	75/15	10.5
SPAS	18	1	65/16	11/7	320/180	5.6
PAIN	16	0	51/12	12/4	183/32	0.0
OMD/OPD	10	2	52/15	5/5	132/75	20.0
LD/WC	9	1	50/14	4/5	175/150	11.1
MEIGE	5	1	60/10	3/2	155/90	20.0
HYPER	4	1	70/20	0/4	200/50	25
GEN DYS	2	0	60/5	0/2	400/100	0.0

MV—mean value; SD—standard deviation; CD—cervical dystonia; HFS—hemifacial spasms; BLE—blepharospasm; SPAS—upper or lower limb spasticity; PAIN—pain syndromes; OMD/OPD—oromandibular or oropharyngeal dystonia, including spasmodic dysarthria; LD/WC—limb dystonia or writer’s cramp; MEIGE—Meige syndrome; HYPER—hyperhidrosis or hypersalivation; GEN DYS—generalized dystonia.

Demographical as well as treatment-related data were also extracted from the charts (see Table 1). The total dose per session and doses per site and the preparation used were documented. All three BoNT/A preparations licensed in Europe were used. For the sake of comparison, doses were transformed into unified dose units, following European consensus recommendations [22]. Doses of incobotulinum A (incoBoNT/A; Xeomin^®^, Merz Pharmaceuticals, Germany) and onabotulinum A (onaBoNT/A; Botox^®^, Allergan, Erwan, CA, USA) remained unchanged; abobotulinum A doses (aboBoNT/A; Dysport^®^, IpsenPharma, Paris, France) were divided by 3.

There was only one patient with cervical dystonia (CD) who developed secondary treatment failure (STF) and was switched to incoBoNT/A [23], without any effect. They were then treated with 12500 U rimabotulinum B (rimaBoNT/B; Soliste, Neuro/MyoBloc^®^, USA). This patient was not included in Table 1 and Table 2.

**Table 2 medicina-58-00935-t002:** Details of the 15 BoNT-ADEVs.

Indication	*n*=	Number of Patients with BoNT-ADEVs	ADOT-BoNT-ADEV
CD	5	Patient 1, with retrocollis and retrocaput, experienced severe neck weakness after injection with only 100 U Dysport^®^. (Figure 1)	ADOT
		Patient 2, with severe retrocaput, experienced difficulties in swallowing after injection of the deep neck muscles with 100 U Xeomin^®^ per side	---
		Patient 3, with laterocollis and head tremor, experienced difficulties in swallowing after injection of the left lateral muscle group with 500 U Dysport^®^.	---
		Patient 4, with laterocollis und torticaput to the left side, experienced neck muscle weakness and muscle pain on the right side after injection of the left splenius and semispinalis capitis muscle with 200 U Xeomin^®^.	ADOT
		Patient 5, with epsilon-glycan-positive dystonia, who was treated because of severe head jerks to the right side, experienced weakness of shoulder elevation after injection of the right splenius capitis and levator scapulae muscle with 200 U Xeomin^®^.	ADOT
HFS	2	Patient 1, with blepharospasm and additional hemifacial spasm on the left side. After injection of 60 U Dysport^®^ into the right orbicularis oculi muscle, 10 U Dysport^®^ per edge of the right eye lid, and 7.5 U Dysport^®^ into the edges of the left eye lid, a moderate ptosis on the left side developed, despite dose reduction on the left side (Figure 2).	ADOT
		Patient 2, with stable scheme and doses of 70 U Dysport on the left eye, reported significant ptosis of the left eye.	ADOT
BLE	2	Patient 1, with unusual combination of myasthenia gravis and blepharospasm, was injected with 30 U Botox^®^ per orbicularis oculi muscle. Approx. 3 weeks after injection, she reported double vision for 2 to 3 weeks.	---
		Patient 2, with tonic and phasic muscle contraction of the orbicularis oculi muscle, was injected with 17.5 U Botox per oo and 5 U per edge of the eye lid, per side. This scheme was used without any complications for years. She reported severe ptosis on both eyes after intensive sun exposure for several hours, approx. 60 min after injection.	ADOT
SPAS	1	Patient with stiff knee gait after stroke, was injected with 500 U per quadriceps muscle. He claimed to have difficulty standing up and climbing stairs.	ADOT
OMD/OPG	2	Patient 1, with jaw opening dystonia. After injection of the pterygoid muscles with 30 U Xeomin^®^, from the outside, a paresis of jaw opening occurred.	ADOT
		Patient 2, with complex tongue movement, had received 3 × 10 U Botox^®^ per side of the tongue. Approx. 3 days later, the patient reported having difficulties in swallowing for 4–5 weeks. The localization of the side effect was difficult to determine.	---
LD/WC	1	Patient with writer´s cramp of extensor type. After 10 U Dysport^®^ into the extensor indicis and 10 U Dysport^®^ into the extensor digitorum communis, muscle severe paresis of the middle finger extensor occurred (Figure 3).	ADOT
MEIGE	1	Patient with involvement of the masseter muscle on both sides, experienced a moderate reduction in bite strength after injection of 30 U Botox^®^ per masseter muscle.	ADOT
HYPER	1	Patient with carcinoma and removal of the submandibular glands, suffered from severe hypersalivation. After treatment with 100 U Xeomin^®^ per parotid gland, he reported having a very dry mouth.	ADOT

ADOT—this ADEV was classified as ADOT-BoNT-ADEV; CD—cervical dystonia; HFS—hemifacial spasms; BLE—blepharospasm; SPAS—upper or lower limb spasticity; OMD/OPD—oromandibular or oropharyngeal dystonia including spasmodic dysarthria; LD/WC—limb dystonia or writer´s cramp; MEIGE—Meige syndrome; HYPER—hyperhidrosis or hypersalivation.

**Figure 1 medicina-58-00935-f001:**
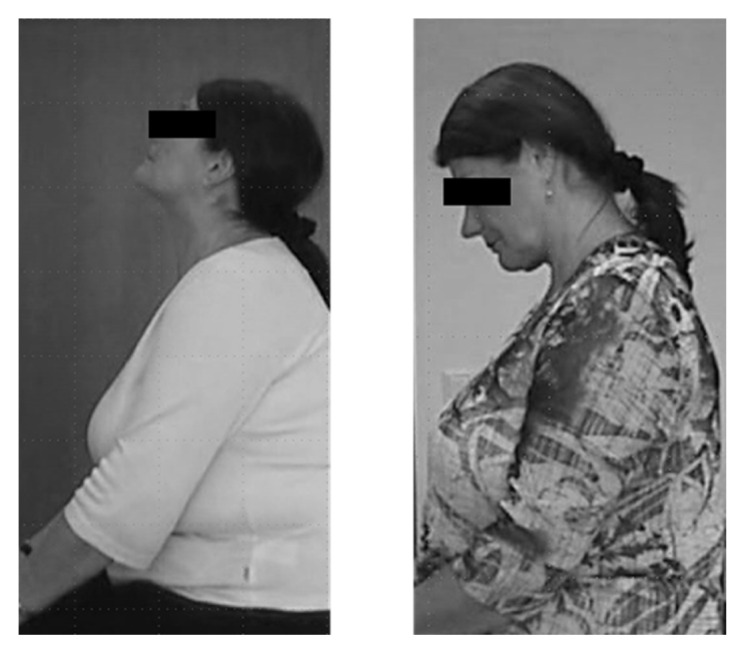
This patient suffers from “retrocollis and retrocaput” (left side). One week after injection of a low dose (100 U) of abobotulinumtoxinA (Dysport^®^), she experienced severe neck weakness and a dropped head.

**Figure 2 medicina-58-00935-f002:**
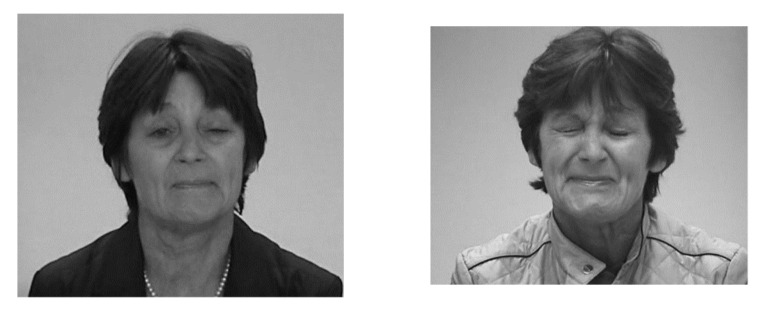
This patient suffers from an extremely rare combination of a blepharospasm with eye- opening dystonia, and an additional hemifacial spasm on the left side. Although the dose used for the left eye was highly reduced to 20% of the dose used for the right eye, a ptosis of the left eye occurred.

**Figure 3 medicina-58-00935-f003:**
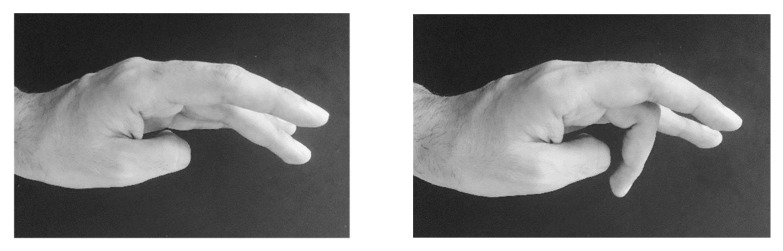
It is well known that the middle finger extensor muscle is supersensitive to BoNT injections. In the BoNT treatment of patients with writer’s cramp of the extensor type, a paresis of the middle finger is a well-known side effect. Left side: before BoNT/A; right side: after BoNT/A.

## 3. Results

### 3.1. Demographical and Treatment-Related Data

The charts of 198 adult patients who received BoNT/A injections between 1 January and 1 April 2021, by HH were screened. One patient, who was treated with rimaBoNT/B, was excluded from further analysis.

Patients were treated for most of the indications for which BoNT/A is licensed (Table 1). The largest subgroup (*n* = 75) included patients with cervical dystonia (CD). This was followed by patients with hemifacial spasms (*n* = 39; HFS); patients with blepharospasm (*n* = 19; BLE); patients with upper or lower limb spasticity (*n* = 18; SPAS); patients with pain syndromes, such as migraines or tension headaches (*n* = 16; PAIN); patients with oromandibular or oropharyngeal dystonia, including spasmodic dysarthria (*n* = 10; OMD/OPD); limb dystonia or writer’s cramp (*n* = 9; LD/WC); patients with Meige syndrome (*n* = 5; MEIGE (Blepharospasm plus facial dystonia)); patients with hyperhidrosis or hypersalivation (*n* = 4; HYPER); and patients with generalized dystonia (*n* = 2; GEN DYS).

The mean total doses per session were slightly higher than recommended in the license descriptions. This was due to the fact that, in comparison to other cohorts of patients, our patients suffered from a high complexity of clinical presentations. More than 60% of these patients were referred from other BoNT centers because of difficulties during their treatment, the dissatisfaction of the patients or the treating physicians, or suspected secondary treatment failure.

### 3.2. ADEVs and BoNT-ADEVs

Eighteen patients (=9.14%) reported ADEVs. Three of these ADEVs were not classified as BoNT-mediated. All three of these ADEVs occurred in patients with CD. One female patient, who already suffered from segmental dystonia (blepharospasm, oromandibular, and cervical dystonia), reported difficulties in swallowing and breathing during walking after receiving the last injection of 750 U aboBoNT/A. Since these difficulties persisted until the end of the injection cycle, we suspect that this was due to the generalization of dystonia, rather than to a systemic side effect. The other two male patients suffered from a complex cervical dystonia and reported difficulty swallowing after receiving the last injection. In both cases, these difficulties persisted for more than 6 weeks and probably indicated a transition from CD to segmental dystonia. In 15 patients (=7.6%), the ADEVs were BoNT-mediated and they are described in detail in Table 2.

The percentage of BoNT-ADEVs varied considerably across the indication subgroups. Because of the small sample size, the BoNT-ADEV percentage was the highest in the HYPER subgroup. The second highest percentage occurred in the OMD/OPD and MEIGE subgroup. No BoNT-ADEVs were reported in the PAIN and GEN DYS subgroup.

The most frequent side effect was neck pain and neck muscle weakness in patients with CD. An example is presented in Figure 1.

The second most frequent side effect was ptosis in the treatment of hemifacial spasms and blepharospasms. An unusual case, with a blepharospasm and an additional hemifacial spasm on the left side is presented in Figure 2.

Third, weakness of the middle finger extensor was demonstrated as an example that shows that a special muscle group may be much more sensitive to BoNT than another muscle group nearby (Figure 3).

Eleven of the fifteen BoNT-ADEVs were classified as ADOT-BoNT-ADEVs. This means that we would have tried to reduce the severity of the BoNT side effects by injecting a locally acting antidote in 5.55% of the patients and in 73% of the BoNT-ADEVs.

## 4. Discussion

### 4.1. Safety and Efficacy of BoNT/A Injections

Treatment with repeated injections of BoNT/A is a generally safe and effective therapy for a variety of indications. Because of its safety and efficacy profile, BoNT/A has received level A recommendations from the American Academy of Neurology (AAN) and the Royal Society of Physicians for a variety of indications [24,25]. The frequency of adverse events depends on the indication (see Table 1 and compare [7,26,27]). In a double-blind study, comparing the ona- and incoBoNT/A treatment of CD [25], 25% to 30% of the patients in both the medication groups experienced ADEVs [26,28]. In a large open-label study that analyzed the safety and efficacy of the first BoNT/A injection in 516 BoNT-naïve CD cases, up to 43% of patients reported ADEVs [29]. Furthermore, in placebo-controlled trials of CD, up to 53% of the placebo-treated patients reported ADEVs [30]. This suggests that a large percentage of reported ADEVs in controlled and open trials were not BoNT-associated.

In a recent review of BoNT/A and BoNT/B treatment of CD, only 21% of the reported ADEVs were BoNT-related [30]. Taking into account this result, the frequency of BoNT-ADEVs appears to be rather low and varies between 5% and 9% in double-blind or open-label studies. Therefore, the overall frequency of BoNT-ADEVs of 7.6% and the frequency of BoNT-ADEVs of 6.7% in the CD subgroup, corresponds with the numbers reported in the literature.

### 4.2. Two Types of BoNT-ADEVs

In general, the frequency of the side effects declines with repeated applications and with the experience of the treating physician. In the present cohort, most patients were referred from other centers because of complications, a lack of efficacy, or the induction of antibodies and side effects. This is the reason why the mean total doses for different indications were rather high.

In clinical practice, BoNT-related side effects can hardly be avoided. Especially in difficult-to-treat cases or in suspected secondary treatment failure (as in many cases from our cohort), side effects occur quite frequently. This results from the treatment of structures (muscles, glands) close to other BoNT-sensitive structures or the use of high doses with the risk of a spread to neighboring structures.

In the present paper, we have distinguished the following two types of BoNT-associated side effects: (i) side effects that result from the overdosing of a special muscle or gland, and (ii) side effects resulting from the spread of BoNT into neighboring structures. In the first case, an antidote treatment (if available) may be considered; in the second case, it is less likely that an improvement can be achieved by antagonizing the side effect with an antidote.

Both types of side effects occur after the uptake and onset of the action of BoNT. A critical question remains, regarding what delay-dependent amount of improvement can be achieved when the antidote is administered post-BoNT injection. This question can only be answered when an antidote is available and specific studies are designed in humans. In cell-based experiments, on the application of copper complexes, the cleavage of SNAP25 could be inhibited in a time-dependent manner, within a short time window of approximately 8 h [20]. When BoNT/A is injected into humans, the uptake of BoNT/A also takes some time. Therefore, it may very well be that, in humans, the time window will be even larger than 8 h.

### 4.3. Combined Use of BoNT Injections and Antidote Application

Much effort is being invested in the development of mutants of BoNT/A, with a shorter duration of action [31]. The use of such mutants is only reasonable in a few applications. If applied repeatedly, the risk of antibody formation is high, as is known from the treatment with BoNT F [32]. Another strategy could be to combine the application of a BoNT preparation with a normal duration and to antagonize the effect of LC activity, if necessary.

The attempt to influence the duration of the action of BoNT/A by the additional application of BoNT/E, which also cleaves SNAP25 but in a different place than BoNT/A, failed to reduce BoNT/A-induced paralysis [33]. However, continuous infusion with 3,4-DP showed a symptomatic reversal of BoNT-induced paralysis and antidotal efficacy after a lethal systemic injection with BoNT/A [18]. Therefore, the systemic application of 3,4-DP may have a beneficial influence on the second type of BoNT/A-induced side effects.

Recent experiments on the reduction in LC activity by means of heavy metal complexes, especially of copper and mercury, also appear to be promising. In cell cultures, as well as in BoNT-treated mice, a reduction in LC activity was demonstrated by the dependence on the delay after the administration of copper complexes, following the BoNT injection [20]. For humans, mercury is too toxic. Copper also has a high cell toxicity, as is known from WD [34,35]. In WD, the amount of free copper is elevated in the blood, but in clinical practice, this elevation is not high enough to reduce the BoNT/A activity significantly [21]. Furthermore, whether copper complexes can be administered in high enough doses to reduce LC activity locally, without toxic effects has still to be shown.

## 5. Conclusions

For the first time, in this study the frequency of BoNT-related side effects was estimated, which can probably be reduced by a locally acting LC antidote (ADOT-BoNT-ADEVs). Such ADEVs seem to occur quite frequently (in approximately 5% of BoNT-treated patients). This underlines the need for the development of such BoNT antidotes in clinical practice, in addition to the need that results from the necessity to treat botulism or a bioterrorist attack.

Therefore, animal experiments are recommended, in which BoNT/A, 3,4-DP, copper complexes, or other substances are locally applicated in different doses and with different delays, to prepare the application of these substances as an antidote in human subjects to reduce the risk of BoNT/A overdosage in clinical practice.

## 6. Limitations of the Study

The estimation of the frequency of the ADOT-BoNT-ADEVs was based on a small sample size (*n* = 197) and on a cohort of patients with a high disease complexity. Nevertheless, we believe that the estimation is realistic since the frequency of BoNT-ADEVs in the present cohort of BoNT-treated patients matches the frequencies of BoNT-ADEVs reported in the literature. Currently, the assumption that a substance can be developed that can effectively reduce the BoNT-LC activity locally is highly speculative.

## Data Availability

Data available on request due to restrictions, e.g., privacy or ethical considerations. The data presented in this study are available on request from the corresponding author.

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
