# Peer review of "The Necessity of a Locally Active Antidote in the Clinical Practice of Botulinum Neurotoxin Therapy: Short Communication"

_medicina, 2022, doi:10.3390/medicina58070935_

Round 1

Reviewer 1 Report

Botulinum neurotoxin A (BoNT/A) is widely used in various medical indications requiring a limitation of cholinergic neuron activity. In this short report, the authors show the prevalence of side effects consecutive to BoNT/A treatment. Although the number of patients included in this study is limited, it is shown that the side effects are more prevalent in certain BoNT applications. Local administration of BoNT inhibitor is proposed to control the emergence of side effects.

Certain side effects are due to BoNT overdosing. The doses administrated in this study are indicated in Table 1. Recommended doses or reference doses from other studies could be also indicated in an additional columns for comparison.

It is mentioned that a part of side effects results from BoNT spreading to neighboring structures. Are these side effects due to difficulty of BoNT administration in certain applications or from the skill and experience of physicians? This would be further commented.

Author Response

Botulinum neurotoxin A (BoNT/A) is widely used in various medical indications requiring a limitation of cholinergic neuron activity. In this short report, the authors show the prevalence of side effects consecutive to BoNT/A treatment. Although the number of patients included in this study is limited, it is shown that the side effects are more prevalent in certain BoNT applications. Local administration of BoNT inhibitor is proposed to control the emergence of side effects.

Certain side effects are due to BoNT overdosing. The doses administrated in this study are indicated in Table 1. Recommended doses or reference doses from other studies could be also indicated in an additional columns for comparison.

It is mentioned that a part of side effects results from BoNT spreading to neighboring structures. Are these side effects due to difficulty of BoNT administration in certain applications or from the skill and experience of physicians? This would be further commented.

Reviewer 1 raises a relevant point: We had only shortly mentioned that BoNT doses were higher than recommended doses. For a more detailed comparison of doses also disease severity has to taken into account. This is now addressed. But we did not add further details of dosing to keep the manuscript short. The different BoNT/A preparations are not licensed for all indications mentioned in our paper. This implies that dosing has to be addressed for each indication and each BoNT/A preparation separately.

This second relevant point raised by reviewer 1 is now discussed in more detail.

We are thankful for these helpful comments.

Reviewer 2 Report

In the short communication titled “The necessity of locally active antidot[e] in clinical practice of botulinum neurotoxin therapy”, Hefter and Samadzadeh argue the need for antidotes to reverse unwanted paralysis resulting from therapeutic botulinum neurotoxin injections. While the metanalysis findings quantifying adverse effects is interesting and provides context to anecdotal data or data scraped from FDA adverse effects database (which is of dubious value), the manuscript suffers from several significant issues that should be addressed. In particular, the authors focus on copper treatment, for which no in vivo data exists, while ignoring more advanced treatments that have been shown to be effective in vivo and hold clear advantages over copper.

Major criticisms:

1.     English can be a difficult language for non-native speakers. This manuscript suffers from many grammatical errors that, at times, interfere with comprehension. I suggest the authors use an English-speaking editor to correct these problems.

2.       It is unclear why the authors focus on copper therapy in part 1 (particularly since there is no correlation to paralytic duration and serum free copper levels AND excess copper can be poisonous), while ignoring other treatments that have proven more effective and are substantiated by a greater number of publications. For example, while the authors briefly mention 3,4-diaminopyridine (DP), which is approved for treatment of Lambert Eaton Myasthenic Syndrome in Europe, Japan and the US and thus available immediately for repurposing, they fail to acknowledge recent data describing the DP mechanism of action in treatment of botulism (PMID: 29366638), showing symptomatic reversal in animal models of local [PMID: 31996484] and antidotal efficacy following lethal systemic challenge with serotype A (the serotype used for most therapeutic neurotoxin formulations) [35659174] . Based on these data, the likelihood exists that 3,4-DAP will effectively reverse partial paralyses (e.g., caused by the authors 2nd type of BoNT-ADEVs) without affecting paralysis in the injected muscle group. Anecdotal studies have shown clinical efficacy after foodborne botulism [PMID: 11984982, 24007658), although dosing and treatment strategies remain uncertain. Separately, recent papers have shown intracellular delivery of LC-blocking antibodies also provide antidotal benefits [PMID: 33408188, 33408184]. These options should be addressed by the authors and considered as potential treatments in the introduction and discussion.

3.       The overall structure of the article is a bit confusing.  I do appreciate the authors summary of the prevalence and nature of adverse effects. However, the authors should focus on improving their stated rationale for these studies/case reviews. For example, the culminating sentence of section 1 is the hook for the remainder of the article. This needs to be written more clearly to present a stronger justification for the meta-analysis. I suggest the authors start by describing the need to understand adverse events (without discussing potential treatments in the introduction), present the adverse effects meta-analysis and interpretations, then finish the communication by addressing potential treatments for the adverse effects in the discussion. This way they do not repeat themselves discussing the copper findings in the introduction and the discussion.

4.       On line 174, the authors should use the term “generally [or usually] safe and effective”; otherwise, (1) there is no need for this review and (b) it seems at odds with ‘BoNT-related side effects can hardly be avoided’ [line 2050).  In general, the authors present an inconsistent argument for the need for an antidote, simultaneously saying side effects are minor and resolve readily over a few weeks, yet also saying there is a  need for a local antidote, which presumably carries its own dose-limiting toxicities. recommend reframing the conclusion to focus less on copper treatment, which, again, has not shown any efficacy in vivo, to speak more generally about treatment options such as copper, DP and intracellular antibodies.

     I should note that while a local injection of copper might not result in systemic toxicity, the authors completely neglect to consider the likelihood of tissue toxicity caused by high local concentrations after injection. 

6.       Finally, the ‘need’ for a *local* antidotal therapy is not strongly justified in the discussion. Furthermore, while the authors assume a systemic treatment will be equally effective in reversing desired and unwanted effects, see my comments in section 2: that assumption may not be true. The authors should be careful not to neglect other treatment modalities, particularly modalities with greater support than local injection of copper.

Author Response

In the short communication titled “The necessity of locally active antidot[e] in clinical practice of botulinum neurotoxin therapy”, Hefter and Samadzadeh argue the need for antidotes to reverse unwanted paralysis resulting from therapeutic botulinum neurotoxin injections. While the metanalysis findings quantifying adverse effects is interesting and provides context to anecdotal data or data scraped from FDA adverse effects database (which is of dubious value), the manuscript suffers from several significant issues that should be addressed. In particular, the authors focus on copper treatment, for which no in vivo data exists, while ignoring more advanced treatments that have been shown to be effective in vivo and hold clear advantages over copper.

Major criticisms:

1.     English can be a difficult language for non-native speakers. This manuscript suffers from many grammatical errors that, at times, interfere with comprehension. I suggest the authors use an English-speaking editor to correct these problems.

2.       It is unclear why the authors focus on copper therapy in part 1 (particularly since there is no correlation to paralytic duration and serum free copper levels AND excess copper can be poisonous), while ignoring other treatments that have proven more effective and are substantiated by a greater number of publications. For example, while the authors briefly mention 3,4-diaminopyridine (DP), which is approved for treatment of Lambert Eaton Myasthenic Syndrome in Europe, Japan and the US and thus available immediately for repurposing, they fail to acknowledge recent data describing the DP mechanism of action in treatment of botulism (PMID: 29366638), showing symptomatic reversal in animal models of local [PMID: 31996484] and antidotal efficacy following lethal systemic challenge with serotype A (the serotype used for most therapeutic neurotoxin formulations) [35659174] . Based on these data, the likelihood exists that 3,4-DAP will effectively reverse partial paralyses (e.g., caused by the authors 2nd type of BoNT-ADEVs) without affecting paralysis in the injected muscle group. Anecdotal studies have shown clinical efficacy after foodborne botulism [PMID: 11984982, 24007658), although dosing and treatment strategies remain uncertain. Separately, recent papers have shown intracellular delivery of LC-blocking antibodies also provide antidotal benefits [PMID: 33408188, 33408184]. These options should be addressed by the authors and considered as potential treatments in the introduction and discussion.

3.       The overall structure of the article is a bit confusing.  I do appreciate the authors summary of the prevalence and nature of adverse effects. However, the authors should focus on improving their stated rationale for these studies/case reviews. For example, the culminating sentence of section 1 is the hook for the remainder of the article. This needs to be written more clearly to present a stronger justification for the meta-analysis. I suggest the authors start by describing the need to understand adverse events (without discussing potential treatments in the introduction), present the adverse effects meta-analysis and interpretations, then finish the communication by addressing potential treatments for the adverse effects in the discussion. This way they do not repeat themselves discussing the copper findings in the introduction and the discussion.

4.       On line 174, the authors should use the term “generally [or usually] safe and effective”; otherwise, (1) there is no need for this review and (b) it seems at odds with ‘BoNT-related side effects can hardly be avoided’ [line 2050).  In general, the authors present an inconsistent argument for the need for an antidote, simultaneously saying side effects are minor and resolve readily over a few weeks, yet also saying there is a need for a local antidote, which presumably carries its own dose-limiting toxicities. recommend reframing the conclusion to focus less on copper treatment, which, again, has not shown any efficacy in vivo, to speak more generally about treatment options such as copper, DP and intracellular antibodies.

     I should note that while a local injection of copper might not result in systemic toxicity, the authors completely neglect to consider the likelihood of tissue toxicity caused by high local concentrations after injection. 

6.       Finally, the ‘need’ for a *local* antidotal therapy is not strongly justified in the discussion. Furthermore, while the authors assume a systemic treatment will be equally effective in reversing desired and unwanted effects, see my comments in section 2: that assumption may not be true. The authors should be careful not to neglect other treatment modalities, particularly modalities with greater support than local injection of copper.

Based on your suggestion, aside from the correction by our native speaker, we sent that to MDPI editing system for further editing.

Reviewer 2 is absolutely right:

The “copper story” is overrepresented in the manuscript. The first version of the manuscript was written as a reaction on the paper by Bremer et al. (Reference 18) which mentions botulism and terroristic attacks as reasons why BoNT antidots should be developed but not clinical overdosing.

Later on, we realized that the problem of BoNT related side effects in clinical practice were of more general interest, but the “copper overload” was still present in the submitted manuscript as picked up by reviewer 2.

We therefore follow the advice of reviewer 2 and have rewritten introduction and discussion. We incorporate the many recommendations of reviewer 2 and think that this has improved the manuscript considerably.

This is done.

This point is explicitly mentioned now.

We hope that the revised manuscript avoids “the copper overload” and is more balanced now.

Round 2

Reviewer 2 Report

The authors have significantly improved the manuscript. The focus is now appropriately placed on the nature of adverse effects and the potential merits of various symptomatic/antidotal treatments to mitigate adverse effects. There is considerable information included in this short report. The authors have satisfied my concerns.